# Molecular Genetic Diversity and Combining Ability for Some Physiological and Agronomic Traits in Rice under Well-Watered and Water-Deficit Conditions

**DOI:** 10.3390/plants11050702

**Published:** 2022-03-05

**Authors:** Raghda M. Sakran, Mohamed I. Ghazy, Medhat Rehan, Abdullah S. Alsohim, Elsayed Mansour

**Affiliations:** 1Rice Research Department, Field Crops Research Institute, Agricultural Research Center, Giza 12619, Egypt; raghdasakran@yahoo.co.uk (R.M.S.); m_ghazy2050@yahoo.com (M.I.G.); 2Department of Plant Production and Protection, College of Agriculture and Veterinary Medicine, Qassim University, Burydah 51452, Saudi Arabia; medhat.rehan@agr.kfs.edu.eg; 3Department of Genetics, College of Agriculture, Kafrelsheikh University, Kafr El-Sheikh 33516, Egypt; 4Agronomy Department, Faculty of Agriculture, Zagazig University, Zagazig 44519, Egypt

**Keywords:** rice, drought stress, heterosis, genetic diversity, principal component analysis, cluster analysis, gene action

## Abstract

Water deficit is a pivotal abiotic stress that detrimentally constrains rice growth and production. Thereupon, the development of high-yielding and drought-tolerant rice genotypes is imperative in order to sustain rice production and ensure global food security. The present study aimed to evaluate diverse exotic and local parental rice genotypes and their corresponding cross combinations under water-deficit versus well-watered conditions, determining general and specific combining ability effects, heterosis, and the gene action controlling important traits through half-diallel analysis. In addition, the research aimed to assess parental genetic distance (GD) employing simple sequence repeat (SSR) markers, and to determine its association with hybrid performance, heterosis, and specific combining ability (SCA) effects. Six diverse rice genotypes (exotic and local) and their 15 F_1_ hybrids were assessed for two years under water-deficit and well-watered conditions. The results revealed that water-deficit stress substantially declined days to heading, plant height, chlorophyll content, relative water content, grain yield, and yield attributes. Contrarily, leaf rolling and the sterility percentage were considerably increased compared to well-watered conditions. Genotypes differed significantly for all the studied characteristics under water-deficit and well-watered conditions. Both additive and non-additive gene actions were involved in governing the inheritance of all the studied traits; however, additive gene action was predominant for most traits. The parental genotypes P_1_ and P_2_ were identified as excellent combiners for earliness and the breeding of short stature genotypes. Moreover, P_3_, P_4_, and P_6_ were identified as excellent combiners to increase grain yield and its attributes under water-deficit conditions. The hybrid combinations; P_1_ × P_4_, P_2_ × P_5_, P_3_ × P_4_, and P_4_ × P_6_ were found to be good specific combiners for grain yield and its contributed traits under water-deficit conditions. The parental genetic distance (GD) ranged from 0.38 to 0.89, with an average of 0.70. It showed lower association with hybrid performance, heterosis, and combining ability effects for all the studied traits. Nevertheless, SCA revealed a significant association with hybrid performance and heterosis, which suggests that SCA is a good predictor for hybrid performance and heterosis under water-deficit conditions. Strong positive relationships were identified between grain yield and each of relative water content, chlorophyll content, number of panicles/plant, number of filled grains/panicle, and 1000-grain weight. This suggests that these traits could be exploited as important indirect selection criteria for improving rice grain yield under water-deficit conditions.

## 1. Introduction

Rice (*Oryza sativa* L.) is a vital cereal food crop for most parts of the world [1,2]. It has valuable nutritional benefits, being high in carbohydrates, low in fat, and rich in minerals, calories, protein, and vitamins [3]. Its acreage is approximately 162 million hectares, producing about 756 million tones [4]. Moreover, this production should be increased to cope with continuing population growth and the threat of environmental pressures [5]. Drought stress is a harsh environmental factor with devastating impacts on global rice production [6,7]. Nearly half the cultivated area of rice worldwide is predicted to suffer from water shortages, causing a substantial reduction in rice yield [8]. Rice requires standing water throughout its growing season for acceptable production. Rice is considered vulnerable to water shortage during its vegetative and reproductive stages, impacting grain yield. Water deficit accelerates flowering and elevates spikelet sterility, which results in fewer grains per panicle. Consequently, it is vulnerable to reduced yields due to water scarcity [9]. Thus, developing drought-tolerant genotypes is imperative in order to maintain global food security [10,11]. Continuous efforts are underway to breed high-yielding and drought-tolerant rice genotypes, particularly in the face of current climate change. 

An effective program for breeding high-yielding and drought-tolerant genotypes must evaluate available materials in order to categorize the appropriate sources of drought tolerance [12]. Therefore, assessing the combining ability and nature of the gene action controlling agronomic traits could assist in identifying suitable parents for crossing, as well as promising recombinants, in order to improve drought tolerance [13,14]. Diallel mating is an efficient method of biometric analysis for the study of general (GCA) and specific (SCA) combining ability effects, enabling the identification of the gene action implicated in several traits in rice [15]. Additionally, it assists in understanding the heterotic effects of the offspring at a preliminary stage of breeding programs [16]. The selection of parental genotypes with good GCA effects and offspring with high SCA effects for eligible traits can be thereby enhanced. Several studies have exploited GCA and SCA to recognize good parents and identify the best crosses for improving stress tolerance in rice [3,17,18,19,20,21]. 

Assessing genetic diversity is an integral aspect of the development of new genotypes with a desired combination of traits [22,23,24]. Moreover, it accelerates the detection of promising genotypes without the need to evaluate all possible cross combinations in breeding programs [25,26]. Morphological and biochemical markers are environmentally influenced, and are labor-intensive and costly to study [27]. Conversely, DNA-based markers are more reliable, stable, and repeatable. Accordingly, DNA-based markers have been widely applied to investigate genetic divergence among genotypes [28,29]. Microsatellites or simple sequence repeats (SSR) are valuable markers due to their co-dominant transmission, multi-allelic nature, relative abundance, genome coverage, informativeness, and low DNA requirement [30]. Numerous published reports have revealed a significant relationship between GD-based molecular markers with F_1_ hybrid performance and heterosis [31,32,33], although other studies have detected no significant correlation [22,26,34,35]. We hypothesized that crossing diverse exotic and local rice genotypes with different levels of tolerance to water deficit would provide promising high-yielding and tolerant F_1_ hybrids. Therefore, the current study aimed to accomplish the following: (i) to explore the performance of six diverse rice genotypes and their 15 F_1_ crosses for some physiological and agronomic traits under well-watered and water-deficit conditions; (ii) to assess the combining ability, heterosis, and gene action type that regulates the inheritance of the studied traits; (iii) to investigate parental genetic distance and its association with hybrid performance, heterosis, and SCA; and (iv) to study the associations among evaluated traits under well-watered and water-deficit conditions. 

## 2. Materials and Methods

### 2.1. Parental Genotypes and Hybridization

Six diverse rice (*Oryza sativa* L.) genotypes were selected according to their distinct origin and tolerance to water deficit. The selected parents included tolerant exotic genotypes and adopted local cultivars that are sensitive to drought stress. Seeds were obtained from genetic stock of the Rice Research and Training Center (RRTC). The pedigree of utilized parents is displayed in Table 1. A half-diallel mating design (6 × 6) excluding reciprocals was employed to produce 15 F_1_ hybrids during the summer season of 2019. The hot-water emasculation method was utilized according to Jodon [36], as modified by Butany [37].

### 2.2. Experimental Site and Agronomic Practices

The parental genotypes and generated F_1_ hybrid combination crosses were assessed during two growing seasons in 2020 and 2021 at the Experimental Farm of Rice Research Department, Sakha Agricultural Research Station, Kafr El-Sheikh, Egypt (31°09′ N, 30°9′ E). The experimental site is hot, with no precipitation events during rice growing seasons, which is representative of the summer season in Egypt. The meteorological data are shown in Appendix A. The physical and chemical soil properties of the experimental site during both seasons are described in Appendix A. The parental genotypes and obtained F_1_ hybrid combinations were assessed under two irrigation regimes; well-watered and water-deficit conditions in separated experiments. A Randomized Complete Block Design (RCBD) with three replications was used in each experiment. The well-watered condition (W) was performed using continuous flooding every 4 days with an adequate depth of submersion that ensured all surface areas were covered by water in each irrigation incident. The water-deficit treatment (S) was performed using irrigation every 12 days without standing water. The stress condition was applied after 15 days from the transplantation date until maturity. The applied irrigation quantities for each treatment were measured using a flow meter, and were 13,100 and 8360 m^3^/ha under well-watered and water-deficit conditions, respectively. The seeds of each genotype (parents and their corresponding F_1_ crosses) were sown in the nursery on 7 and 5 of May in the 2020 and 2021 seasons, respectively, then transplanted to the field after 30 days. The seedlings of each genotype (the parents and F_1s_) were individually transplanted in three rows per replicate. Each row was 4.0 m long with a spacing of 20 × 20 cm among rows and hills. Nitrogen fertilizer at a rate of 165 kg N ha^−1^ was applied in three splits in the form of urea (46.0% N). Phosphorous was applied at a rate of 37 kg P_2_O_5_ ha^−1^ as super-phosphate (15% P_2_O_5_), and potassium at a rate of 50 kg K_2_O kg/ha as potassium sulfate (48% K_2_O). Zinc fertilizer was applied at a rate of 24 kg/ha ZnSO_4_. Other standard agricultural practices such as weed control and disease protection were applied.

### 2.3. Measured Traits

#### 2.3.1. Morphological and Physiological Traits

The value of days to heading (DTH) was measured as the number of days from transplanting to the date when 50% of panicles were fully exerted in each plot. The plant height (PH; cm) was recorded as the distance from the soil surface to the tip of the main panicle at maturity. Chlorophyll content (CHLC; SPAD unit) was recorded utilizing a SPAD meter (Model: SPAD-502, Hangzhou Mindfull Technology Co., Ltd., Hangzhou, China) from the topmost completely expanded leaves on the main panicle during the flowering period. Leaf rolling (LR) was recorded using visual assessment following the method outlined by De Datta et al. [38]. Relative water content (RWC) was measured as proposed by Barrs and Weatherley [39] using the following equation: RWC = ((FW − DW)/(TW − DW)) × 100, where FW is fresh leaf weight, DW is leaf dry weight, and TW is turgid leaf weight. 

#### 2.3.2. Grain Yield and Its Related Traits

At harvest, the number of panicles per plant (NP) was measured by counting the number of panicles/plant of 10 randomly selected plants in each plot. The number of filled grains per panicle (NFG) was recorded by separating and counting the filled grains of 10 main panicles selected randomly from each plot. The 1000-grain weight (TGW; g) was calculated based on the weight of 1000 grains sampled from each plot. The sterility percentage (SP, %) was recorded by dividing the number of unfilled grains by the total grains from 10 panicles/plot. Grain yield/plant (GYPP; g) was recorded as the weight of the individual plant grain yield and adjusted to 14% grain moisture content.

### 2.4. Molecular Analysis

Genomic DNA was extracted utilizing the CTAB method [40] from a healthful portion of leaves obtained from 20-day-old seedlings. Twenty-nine SSR markers related to drought tolerance traits/QTLs were applied [41,42,43,44]. Only 16 polymorphic markers were detected and accordingly were employed for SSR analysis. The sequence information of primers used is presented in Appendix A, and chromosomal locations and repeat motifs are provided on the Gramene website (http://www.gramene.org, accessed on 26 January 2022). PCR analysis was performed utilizing a 10 μL reaction volume including 1 Taq DNA polymerase unit, 0.2 mM dNTPs, 2 mM MgCl_2_, 0.5 μM reverse and forward primers, and 1 μL of 20 ng/μL genomic DNA template. The PCR reaction was initially started by denaturation at 94 °C for 2 min followed by 94 °C for 30 s, annealing at 55–64 °C (based on primer Tm) for 30 s, extension at 72 °C for 30 s for thirty-five cycles, and the program was completed with a final extension step at 72 °C for 3 min. Amplification products were separated using a gel electrophoresis unit incorporating 2% agarose gel, stained with ethidium bromide and visualized under a UV-Gel documentation system. Allele number, major allele frequency, gene diversity, and the polymorphic information content (PIC) were identified for all markers utilizing Power-Marker version 3.25. Genetic distances were estimated using the PAST program. The dendrogram tree was constructed with the unweighted pair group method utilizing arithmetic averages (UPGMA) within the computational package MVSP version 3.1.

### 2.5. Statistical Analysis

The analysis of variance (ANOVA) was performed for all measured traits using R statistical software version 4.1.1 The combined analysis was applied whenever the homogeneity test was non-significant using Bartlett’s test. Combining ability analysis was applied as outlined by Griffing’s method 2, model 1 [16]. Heterosis was computed as follows: mid-parent (MP) heterosis = (F_1_ − MP)/MP × 100 and better-parent (BP) heterosis = (F_1_ − BP)/BP × 100. 

Four tolerance indices were estimated to determine potentially drought-tolerant genotypes as follows: Mean productivity (MP)$=\frac{Y_{s}+{\;Y}_{p}}{2}$, [45]

Geometric mean productivity (GMP) $=\sqrt{Y_{s}\times Y_{p}}$, [46]

Stress tolerance index (STI) $\mathrm{STI}=\frac{Y_{s}\times Y_{p}}{{\left({\overset{-}{Y}}_{p}\right)}^{2}}$, [46]

Yield index (YI) $=\frac{Y_{s}}{{\overset{-}{Y}}_{s}}$, [47]
where Y_p_ is yield of each genotype under well-watered conditions, Y_s_ is yield of each genotype under water-deficit conditions, and Ȳ_p_ and Ȳ_s_ are the means of all genotypes under well-watered and stress conditions, respectively. The hierarchical cluster analysis was applied based on tolerance indices to differentiate the evaluated genotypes based on their drought tolerance following the method of Ward Jr [48]. Principal component analysis (PCA) was performed to evaluate associations among the studied traits.

## 3. Results

### 3.1. Analysis of Variance

The analysis of variance results indicated significant impacts of irrigation (I), genotype (G), and their interactions (G × I) on all of the studied traits (Table 2). Dividing the genotypic effect into GCA and SCA components revealed that the GCA and SCA mean squares were highly significant for all assessed characteristics. The interaction effects of GCA × I and SCA × I were significant for all evaluated traits. The ratio of GCA/SCA was greater than the unity for all considered traits, except chlorophyll content (CHLC), number of filled grains/panicle (NFG), sterility percentage (SP), and 1000-grain weight (TGW). Furthermore, the extent of GCA × I interaction was superior to SCA × I for all characteristics, except PH, CHLC, NP, and TGW.

### 3.2. Mean Performance of Parents and F_1_ Hybrids

The assessed rice parental genotypes and their F_1_ hybrids manifested a wide variation for all measured traits under both well-watered and stress conditions. Water-deficit stress caused early heading in all tested genotypes compared with normal irrigation. The mean values of days to heading (DTH) reduced from 103.1 to 99.3 days due to water deficiency (Figure 1A). The parents P_3_, P_1_, and P_2_ and the hybrids P_1_ × P_2_, P_1_ × P_3_, and P_2_ × P_6_ displayed the earliest heading, whereas the latest heading was exhibited by the parent P_6_ and the hybrids P_1_ × P_6_, P_2_ × P_4_, P_3_ × P_5_, and P_5_ × P_6_ under both normal and water-deficit conditions (Figure 2A). Plant height (PH) declined on average from 118.4 to 103.0 cm, as the amount of irrigation decreased (Figure 1B). The parents P_3_ and P_2_ and hybrids P_1_ × P_5_, P_1_ × P_3_, and P_2_ × P_3_ produced the shortest plants, whereas the tallest ones were from the parents P_6_, P_4_ and the hybrids P_3_ × P_4_, and P_4_ × P_6_ under both irrigation regimes (Figure 2B). Water-deficit treatment also caused a considerable increase in leaf rolling (LR), which rose on average from 1.87 under well-watered conditions to 4.59 under water-deficit conditions (Figure 1C). The parental genotypes P_2_, P_3,_ and P_6_ and the hybrids P_1_ × P_3_, P_1_ × P_6_, and P_2_ × P_6_ showed the lowest values under normal conditions, while the parents P_3_, P_5_ and P_6_ and the hybrid combinations P_3_ × P_5_, P_3_ × P_4_, and P_4_ × P_6_ had the lowest values under stress conditions (Figure 2C). Relative water content (RWC) also was negatively affected by water limitation; it decreased from 84.34% under regular irrigation to 72.04% under water-deficit conditions (Figure 1D). The highest values were exhibited by the parents P_4_ and P_6_ and the hybrids P_1_ × P_3_, P_3_ × P_4_, and P_4_ × P_6_ under both treatments (Figure 2D). Likewise, water deficiency decreased chlorophyll content (CHLC) from 43.83 to 39.37 (Figure 1E). The highest values for chlorophyll content were recorded for the parents P_2_ and P_6_ and the hybrids P_1_ × P_2_, P_1_ × P_4_, P_1_ × P_6_, and P_3_ × P_4_ under both conditions (Figure 2E). Under water-deficit conditions, the number of panicles per plant (NP) also declined from 20.13 to 15.52 (Figure 1F). The parental genotypes P_3_ and P_4_ had the highest mean values under both conditions (Figure 3A). The cross combinations P_1_ × P_2_, P_1_ × P_3_, P_2_ × P_6_, P_3_ × P_4_, P_4_ × P_5_, and P_4_ × P_6_ produced the highest number of panicles per plant under well-watered conditions, whilst P_3_ × P_4_, P_4_ × P_6_, and P_3_ × P_6_ produced the highest number under stress conditions. In the same manner, the number of filled grains per panicle (NFG) significantly decreased as irrigation decreased, falling from 131.95 to 104.77 (Figure 1G). The uppermost values were exhibited by the parents P_2_ and P_3_ and the hybrid combinations P_1_ × P_3_, P_2_ × P_5_, P_3_ × P_4_, and P_3_ × P_5_ under well-watered conditions (Figure 3B). Under stress conditions, the highest NFG values were displayed by the parents P_3_ and P_4_ and the hybrids P_1_ × P_3_, P_2_ × P_5_, P_3_ × P_4_, P_4_ × P_5_, and P_4_ × P_6_. On the other hand, water-deficit treatment caused a significant increase in the sterility percentage (SP) from 11.83% to 21.04% (Figure 1H). Under well-watered conditions, the genotypes P_1_, P_2_, P_3_, P_1_ × P_2_, P_1_ × P_3_, P_3_ × P_4_, and P_3_ × P_5_ demonstrated the lowest SP, while P_3_, P_4_, P_6_, P_2_ × P_6_, P_3_ × P_4_, and P_4_ × P_6_ recorded the lowest SP under water-deficit conditions (Figure 3C). Water-deficit treatment reduced the 1000-grain weight (TGW) from 26.81 to 23.35 g (Figure 1I). The heaviest TGW was recorded for the parental genotypes P_1_, P_2_, and P_6_ and the hybrids P_1_ × P_2_, P_1_ × P_3_, P_1_ × P_6_, P_3_ × P_4_, and P_3_ × P_5_ under well-watered conditions. The parents P_5_ and P_6_ and the hybrids, P_1_ × P_3_, P_2_ × P_6_, P_3_ × P_4_, P_3_ × P_6_, and P_4_ × P_6_, on the other hand, had the highest TGW values under water-deficit conditions (Figure 3D). Finally, grain yield per plant (GYPP) reduced significantly, from 38.39 to 27.33 g under well-watered and stress conditions, respectively (Figure 1J). The parents P_1_, P_2_, and P_3_ and the hybrids P_1_ × P_2_, P_1_ × P_4_, P_2_ × P_5_, and P_3_ × P_4_ exhibited the highest grain yield under well-watered conditions, while the parents P_3_, P_4_, and P_6_ and the hybrids P_1_ × P_4_, P_3_ × P_4_, P_3_ × P_6_, and P_4_ × P_6_ produced the highest grain yield under stress conditions (Figure 3E).

### 3.3. Genotypic Classification According to Drought Tolerance Indices

Drought tolerance indices were estimated for the assessed rice parents and their corresponding hybrids. These indices were employed to categorize the evaluated genotypes based on their drought-stress tolerance. The assessed genotypes were categorized into three groups with distinct genotypes (Figure 4). The genotypes in group A included six genotypes, specifically P_3_, P_4_, P_1_ × P_4_, P_3_ × P_4_, P_3_ × P_6_, and P_4_ × P_6_, which possessed superior values for tolerance indices; hence, they are considered drought-tolerant genotypes. Group B, on the other hand, consists of eight genotypes with intermediate values for the tolerance indices; consequently, they are categorized as moderately drought-tolerant genotypes. However, the seven genotypes in groups C displayed the lowest values of tolerance indices; hence, they are deemed drought-sensitive genotypes.

### 3.4. General Combining Ability (GCA) Effects

The parents with significant and positive GCA effects are vital for all studied traits, except DTH, PH, LR and SP where negative values are desirable. The GCA effects for the assessed traits varied greatly among the evaluated parents (Table 3). The parental genotypes P_1_, P_2_, and P_3_ exhibited the highest significant and negative GCA effects for DTH and PH under both well-watered and stress conditions. Similarly, the highest significant and negative GCA effects for LR were recorded by P_3_, P_4_, and P_6_ under water-deficit conditions. Otherwise, significant and positive GCA effects for RWC were observed for P_3_ under drought stress conditions, and for P_4_ and P_6_ under both conditions. In addition, positive and significant GCA effects for CHLC were exhibited by P_1_, P_2_, and P_6_ under both conditions. The parents P_3_ and P_4_ recorded the highest significant and positive GCA effects for NP and NFG. The best combiners for SP were demonstrated by P_1_ and P_2_, under well-watered conditions; P_4_ and P_6_, under water-stress conditions; and P_3_ under both conditions. Meanwhile, the largest positive GCA effect for TGW was observed for P_1_ under well-watered conditions and P_6_ under water-deficit conditions. The highest positive and significant GCA for grain yield was obtained by P_1_ and P_2_ under well-watered conditions, P_6_ under water-stress conditions, and P_3_ and P_4_ under both conditions, demonstrating that they are appropriate candidates for GY improvement. 

### 3.5. Specific Combining Ability (SCA) Effects

The hybrids exhibited considerable variation regarding the SCA effects for all the studied traits (Table 4). Negative and significant SCA estimates for DTH were exhibited by the hybrids P_1_ × P_5_, P_2_ × P_6_, P_4_ × P_5_, and P_4_ × P_6_ under both conditions (Table 4). The crosses P_1_ × P_5_, P_2_ × P_6_, P_3_ × P_6_, and P_4_ × P_5_ displayed significant and negative SCA values for PH across both conditions. The highest negative and significant SCA effects for LR were observed by the crosses P_1_ × P_3_, P_1_ × P_4_, P_1_ × P_6_, P_2_ × P_5_, P_2_ × P_6_, P_3_ × P_4_, P_3_ × P_5_, and P_4_ × P_6_ under water-deficit conditions. Conversely, the uppermost positive and significant SCA estimates for RWC were recorded by P_1_ × P_2_, P_1_ × P_3_, P_2_ × P_5_, P_3_ × P_4_, P_3_ × P_5_, and P_4_ × P_6_ under both conditions. Regarding CHLC, the SCA effects were positive and significant for the crosses P_1_ × P_6_ and P_4_ × P_5_ under normal conditions, for P_4_ × P_6_ under stress conditions, and for P_1_ × P_2_, P_1_ × P_4_, P_2_ × P_5_, and P_3_ × P_4_ under both conditions. Similarly, P_1_ × P_2_, P_1_ × P_3_, P_1_ × P_5_, P_2_ × P_5_, P_3_ × P_4_, and P_4_ × P_6_ expressed the largest significant and positive SCA values for NP. High positive and significant SCA estimates for NFG were exhibited by P_1_ × P_2_, P_1_ × P_3_, P_2_ × P_5_, P_2_ × P_6_, P_4_ × P_5_, and P_4_ × P_6_ under both conditions. The hybrids P_1_ × P_2_, P_3_ × P_4_, P_3_ × P_5_, and P_4_ × P_6_ were identified as good specific combiners for SP. Regarding TKW, the highest positive SCA values were observed for the hybrids P_1_ × P_2_, P_1_ × P_6_, and P_3_ × P_5_ under well-watered conditions, for P_2_ × P_6_ and P_4_ × P_6_ under stress conditions, and for P_1_ × P_3_, P_2_ × P_5_, P_3_ × P_4_, and P_4_ × P_5_ under both conditions. Additionally, the highest positive SCA effects for GYPP were observed for P_3_ × P_5_ under well-watered conditions, P_1_ × P_5_, P_2_ × P_6_, P_3_ × P_6_, and P_4_ × P_6_ under water-deficit conditions, and P_1_ × P_2_, P_1_ × P_4_, P_2_ × P_5_, and P_3_ × P_4_ under both conditions. 

### 3.6. Heterosis Relative to Mid-Parent (MP) and Better-Parent (BP)

The heterosis percentages relative to MP or BP under well-watered and water-deficit conditions are presented in Table 5. Most crosses demonstrated substantial heterosis relative to MP or BP under both conditions. The largest negative MP heterosis for DTH towards earliness was observed for P_1_ × P_5_, P_2_ × P_6_, P_4_ × P_5_, and P_4_ × P_6_ under both conditions, whilst the highest negative BP heterosis was exhibited by P_1_ × P_5_ and P_4_ × P_5_. The cross combinations P_1_ × P_5_, P_1_ × P_6_, P_2_ × P_6_, P_3_ × P_6_, P_4_ × P_5_, and P_5_ × P_6_ manifested significant negative MP heterosis for PH towards shortness. Meanwhile, only the hybrid P_1_ × P_5_ had significant negative BP heterosis for this trait under both conditions. Under drought stress conditions, the hybrids P_1_ × P_3_, P_1_ × P_4_, P_1_ × P_6_, and P_2_ × P_6_ displayed the highest negative and significant MP heterosis for LR, whereases the hybrid P_3_ × P_4_ showed the highest heterosis over BP. Meanwhile, the highest positive and significant MP heterotic effects for RWC were recorded for P_1_ × P_2_, P_1_ × P_3_, P_2_ × P_5_, P_3_ × P_4_, and P_3_ × P_5_ under both treatments, whereas the maximum significant and positive BP heterosis was recorded for P_1_ × P_3_ under well-watered conditions, for P_2_ × P_5_ under water-stress conditions, and for P_1_ × P_2_ and P_3_ × P_4_ under both conditions.

The uppermost positive MP and BP heterotic effects for CHLC were exhibited by the hybrids P_1_ × P_4_ and P_3_ × P_4_ under both conditions. Additionally, positive and significant heterosis over the BP was observed for the hybrid P_1_ × P_6_ under well-watered conditions and for P_4_ × P_6_ under water-stress conditions. With respect to NP, the hybrids P_1_ × P_2_, P_1_ × P_3_, P_1_ × P_5_, P_3_ × P_4_, and P_4_ × P_6_ displayed the largest significant and positive heterotic effects relative to MP and BP under both conditions. The crosses with the best positive MP values for NFG were P_1_ × P_3_, P_3_ × P_4_, and P_3_ × P_5_ under well-watered conditions, P_1_ × P_4_ under water-deficit conditions, and P_2_ × P_5_, P_2_ × P_6_, P_4_ × P_5_, and P_4_ × P_6_ under both conditions. In addition, positive and significant heterosis relative to BP was exhibited by the crosses P_2_ × P_6_ and P_4_ × P_6_ under stress conditions and P_2_ × P_5_ and P_4_ × P_5_ under both conditions. The three hybrids P_1_ × P_2_, P_2_ × P_6_ and P_4_ × P_6_ possessed significantly negative MP heterosis for the SP, while the highest negative and significant BP heterosis was exhibited by the hybrid P_4_ × P_6_ under stress conditions. Concerning TKW, significant and positive MP heterotic effects were observed for the hybrids P_1_ × P_2_, P_1_ × P_6_, P_2_ × P_4_, P_3_ × P_5_, and P_4_ × P_5_ under well-watered conditions and for P_1_ × P_3_, P_2_ × P_5_, and P_3_ × P_4_ under both conditions. Furthermore, significant and positive BP heterotic effects were exhibited by the hybrid P_1_ × P_6_ under well-watered conditions, by P_1_ × P_3_ under stress conditions, and by P_3_ × P_4_ under both conditions. The hybrid combinations P_1_ × P_2_, P_1_ × P_4_, and P_3_ × P_4_ displayed the uppermost positive and significant MP and BP heterotic effects under both conditions for GYPP. Meanwhile, the hybrids P_2_ × P_5_, P_2_ × P_6_, and P_4_ × P_6_ exhibited significantly positive BP heterosis under water-deficit conditions.

### 3.7. Microsatellites Based Polymorphism 

Out of the twenty-nine SSR primers assessed, sixteen polymorphic markers were found throughout the rice genome. The polymorphic SSR markers provided a total of fifty-three alleles, which were utilized to analyze the genetic diversity among the investigated parental genotypes. The allele numbers per locus varied from two (RM315 and RM543) to five (RM20A), with an average of 3.31 alleles/locus (Table 6). The effective number of alleles per locus varied from 1.80 to 3.79 alleles with an average of 2.77 alleles. The maximum effective number of alleles per locus (3.79) was identified for RM20A and RM279. The major allele frequency had an average of 0.49 with a range from 0.33 to 0.67. The gene diversity or heterozygosity (HE) varied from 0.44 (RM543) to 0.74 (RM20A and RM279) with an average of 0.62. Moreover, the average polymorphic information content (PIC) was 0.55 with a range of 0.35 (RM543) to 0.69 (RM20A and RM279). The highly polymorphic SSR markers are displayed in Figure 5.

### 3.8. Genetic Distance (GD) and Cluster Analysis 

Genetic distance based on SSR markers varied from 0.38 to 0.89 with an average of 0.70 (Table 7). The smallest genetic distance (0.38) was detected between the two parents P_1_ and P_2_, whereas the uppermost genetic distance was observed between P_1_ and P_5_ (0.89), followed by P_1_ and P_4_ (0.88). The dendrogram constructed based on GD divided the parental genotypes into two major clusters with internal sub-clusters within both groups (Figure 6). Cluster I comprised the three exotic genotypes, which were further separated into two sub-clusters. The first sub-cluster included P_5_ and P_6_, while the second sub-cluster had only P_4_. Cluster II contained the three Egyptian rice cultivars, which could be further separated into two subgroups; the parental genotypes P_1_ and P_2_ constituted the first subgroup, while P_3_ formed the second one.

### 3.9. Correlation between Parental GD, Hybrid Performance, SCA, and Heterosis

The associations between SSR marker-based GD and F_1_ hybrid performance were not significant for any of the evaluated traits under either the well-watered or water-deficit conditions (Table 8). Similarly, GD displayed a non-significant association with SCA effects, MP heterosis, and BP heterosis. Conversely, the SCA effects of the assessed crosses exhibited a strong correlation with each of the F_1_ hybrids’ performance, MP heterosis, and BP heterosis for all the evaluated traits. Moreover, a highly significant association was detected between F_1_ performance, MP heterosis, and BP heterosis for all the measured traits under both conditions.

### 3.10. Interrelationship among the Studied Traits

Principal component analysis was performed to assess the relationship among evaluated traits under water-deficit conditions. The first two PCAs described most of the variability, accounting for 85.09% (66.76% and 18.33% by PCA1 and PCA2, respectively). Accordingly, the two PCAs were employed to perform the PC-biplot (Figure 7). A strong positive relationship was detected between grain yield and each of relative water content, chlorophyll content, number of panicles per plant, number of filled grains per panicle, and 1000-grain weight. Conversely, a negative association was determined between yield traits and both sterility percentage and leaf rolling. Similar results are depicted by the correlation heatmap, as shown in Appendix A.

## 4. Discussion

Developing high-yielding and drought-tolerant rice genotypes has come to be more imperative to sustain rice production in the face of ongoing population growth and the threats of climate change. The significant genotypic difference detected for all measured traits indicated the presence of adequate genetic variability among the evaluated genotypes. Accordingly, the genetic variations that were discovered could be exploited in establishing a successful rice program for breeding high-yielding and drought-tolerant genotypes. In keeping with these findings, high genetic variations have also been observed for different agronomic traits in rice under well-watered and water-deficit conditions by Venuprasad et al. [49], Monkham et al. [50], Khan et al. [2], and Anusha et al. [17]. The significant difference between tested irrigation regimes resulted in dissimilar impacts on the evaluated genotypes. Moreover, the significant G × I interaction obtained for most of the measured traits revealed differential performances of the genotypes under well-watered and water shortage conditions [51]. Drought is a pivotal environmental stress that detrimentally constraints rice growth and production [52,53]. The obtained results revealed that water deficit caused considerable reductions in all the evaluated characteristics compared to the well-watered condition, except for leaf rolling and the sterility percentage, which significantly increased. These findings are in consonance with prior published reports that confirmed the influence of water limitation on different traits of rice [6,9,11,54]. The obtained results revealed that water-deficit stress induced a reduction in the number of days to heading. Earliness can be considered an escape approach and a resilient adaptation under water-stress conditions [55,56,57]. Moreover, water-deficit stress treatment caused a major decrease in plant height, which may be a result of reduced cell turgor and a constraining of cell division and cell expansion [58,59]. Relative water content (RWC) is a physiological trait that has great importance for evaluating rice genotypes for drought tolerance [60,61]. A drought stress-induced reduction in RWC may be the result of a decline in water uptake from soil to the root system and reduced internal water flow from the root to the leaves [62]. Correspondingly, the decline in chlorophyll content might be due to the destructive impact of the water deficit on photosynthetic enzyme activity and reactive oxygen species accumulation, which affect the chloroplast, resulting in a reduction of carbon assimilation and decreased chlorophyll content, as well as inhibition of the photosynthetic process [63,64]. Yield contributing traits are the eventual products of physiological developments. The notable declines in yield traits under water-deficit stress may be a consequence of a deficiency of absorbed water, along with the inhibition of cell elongation and cell division, delayed cellular growth, and a reduced photosynthetic rate [65,66,67]. Furthermore, water deficit limits assimilate mobilization from leaves and stems to grains, which results in smaller or shriveled grains [68]. Moreover, the reduction in grain number/panicle could be a result of increased pollen inviability [69], spikelet sterility [70], or abortion of immature embryos [49,71]. 

An efficient breeding program relies on the careful choice of parents. The GCA effects are of crucial importance in selecting potential parents that could be exploited in developing improved populations [16], as parental genotypes with the desired GCA transfer additive genes to their offspring, making them useful for breeding programs [45]. The obtained results revealed that GCA values varied between the well-watered and watered deficit conditions, confirming environmental impacts on GCA effects. The parental genotypes P_1_, P_2_, and P_3_ were recognized as good combiners for earliness and shortened plant height under both conditions. This indicates that these parents could be useful in developing early and dwarf genotypes, which are preferred to escape from terminal drought stress [18]. The parents P_1_ and P_6_ were recognized as good combiners for increasing chlorophyll content. Meanwhile, improvements in RWC could be achieved by using the parents P_3_ and P_4_, which had steadily positive and significant GCA estimates under stress conditions. The local parental genotype P_3_ and the two exotic genotypes P_4_ and P_6_ could be favorable combiners for grain yield and some of its contributed traits. Consequently, these parents could transfer their promising alleles to progenies and enhance grain yield under water-stress conditions. Previous studies have also emphasized the significance of using exotic and local parents with positive and high GCA effects for boosting grain yield and related agronomic traits under normal and water-deficit conditions [3,17,21,72].

SCA elucidates the deviation of hybrids’ performance from that of the parents utilized, and allows for estimation of non-additive gene effects. Hybrids with significant SCA effects are great choices for the selection of transgressive segregates. SCA effects revealed that all of the developed hybrids displayed significant SCA in a preferable direction for at least one trait. On the basis of these results, the hybrid combinations P_1_ × P_4_, P_2_ × P_5_, P_3_ × P_4_, and P_4_ × P_6_ can be considered excellent specific combiners for developing high-yielding hybrids under water-deficit conditions. These hybrids were generated from parents with good × good or good × poor general combiners. These results reveal the role of accumulative impacts of additive × additive interactions of positive alleles [73]. Similarly, El-Mowafi, et al. [3] determined that the existence of at least one good general combiner is crucial for ensuring good specific cross combinations. Evidently, none of the evaluated cross combinations recorded significant SCA estimates for all assessed traits. Nevertheless, the hybrid P_3_ × P_4_ was a good specific combiner for CHLC, RWC, NP, NFG, SP, and GYPP under both conditions. Moreover, P_4_ × P_6_ combined well for DTH, RWC, NP, NFG, SP, and GYPP under both conditions. Remarkably, the crosses possessed significant and positive SCA effects and better parent heterotic impacts for grain yield and contributing traits. Consequently, desirable segregants could be projected from these cross combinations. Thus, these crosses could be efficiently employed in rice programs to improve these traits under well-watered and drought stress conditions. 

The significant GCA and SCA effects for grain yield and other evaluated traits indicate that both additive and non-additive gene actions play integral roles in regulating the inheritance of all the assessed traits. However, the variance due to GCA demonstrated a higher magnitude than SCA for most of the studied traits. This implies a greater role for additive gene effects in regulating the inheritance of these traits, and consequently, the recurrent selection method could be efficacious for improving these traits. The obtained results coincide with those of Suvi et al. [18], Zewdu [74], Gramaje et al. [22], and El-Mowafi et al. [3]. They disclosed that the additive gene effects contribute considerably to the inheritance of grain yield and several related traits in rice. On the contrary, other reports have found that non-additive gene action has a major role in the inheritance of agronomic traits in rice [17,19,75]. The significant GCA × I and SCA × I interactions for grain yield and most of the other studied traits indicate the differential performance of the parental genotypes and hybrids under distinct conditions. This demonstrates the necessity of identifying adapted parents and hybrids in each condition individually [76]. The preponderance of GCA × I interaction over SCA × I for most of the studied traits signifies that the additive genetic effects were more affected by the environment than the non-additive ones.

Applying tolerance indices provides valuable information to support the classification of the genotypes based on their tolerance to abiotic stresses. In the present study, cluster analysis was utilized to differentiate drought-tolerant and drought-sensitive genotypes based on tolerance indices. The analysis grouped the assessed genotypes into three clusters (A–C), varying from tolerant to sensitive genotypes. The genotypes P_3_, P_4_, P_1_ × P_4_, P_3_ × P_4_, P_3_ × P_6_, and P_4_ × P_6_ were determined to be drought-tolerant (Figure 4). Consequently, these genotypes could be employed in rice breeding programs to improve grain yield under drought stress conditions. Numerous prior studies have employed tolerance indices and cluster analysis to categorize rice genotypes under water-deficit conditions [65,77].

The SSR markers utilized in the current study reflected the degree of genetic diversity among the assessed parental genotypes. The results indicate that the number of alleles per locus varied from 2 to 5 with an average of 3.31 alleles per locus, a finding that is comparable to Khan et al. [2], Pradhan et al. [78], and Singh et al. [79], which determined values of 3, 3.15 and 3.11 allele/locus, respectively. However, Rahman et al. [80], Das et al. [81], and Tabkhkar et al. [65] identified 4.18, 4.91, and 6.21 alleles/locus, respectively. The variation in allele numbers might be ascribed to the genetic architecture of the genotypes and to the SSR utilized. The most frequent allele possessed an average of 0.49 signifying that 49.0% of the evaluated genotypes had a common allele at any of the examined loci. The PIC measures allelic diversity at a locus; it was relatively high (0.55), which demonstrates good preferential power of the applied markers [19]. The mean PIC value detected in the present study was close to those reported by Zhang et al. [82], Salem and Sallam [83], and Verma and Srivastava [19], who detected an average of 0.54, 57, and 0.51, respectively. Furthermore, the markers RM279 and RM20A displayed higher distinguishing power to differentiate genotypes owing to their high PIC values (0.69). These markers could be utilized for upcoming genetic and mapping analyses in rice [2]. The average genetic diversity that existed among parental genotypes was relatively high (0.70), signifying the existence of considerable genetic diversity based on SSR analysis. The lowest genetic distance was observed between the two Egyptian rice cultivars P_1_ and P_2_, which are japonica-type and drought-susceptible genotypes. On the other hand, the greatest genetic distance was observed between the local parental cultivar P_1_, which is drought-susceptible and japonica-type, and the exotic genotype P_5_, which is an indica type and drought tolerant. Similarly, Chakravarthi and Naravaneni [84] discovered minimal similarity between japonica-type and indica-type lines. On the other hand, Kanawapee et al. [85] determined a moderately high level of similarity between related lines. The microsatellite markers facilitated the grouping of the parental genotypes into two main clusters. Genotypes that share common ancestors were grouped into the same cluster, displaying the efficacy of SSR markers in investigating genetic diversity in rice, as elucidated in previous reports [31,86,87]. The results of the field evaluation indicated that crossing parents from different clusters could result in superior hybrids. Notably, the hybrid P_3_ × P_4_ gave higher grain yield under well-watered and drought stress conditions. It was derived from crossing between group 1 (exotic genotypes) and group 2 (local genotypes). This proves there is the potential to attain more efficient hybrids by crossing diverse genotypes from distinct groups. Thus, parental selection using DNA markers could help in developing superior rice hybrids and population improvements [87], a finding with valuable implications for hybrid rice programs. The non-significant correlation recorded between SSR-based GD, hybrid performance, heterosis (mid or better parent) and SCA effects for all the studied traits could be a result of using a relatively small number of markers. Analogous findings were reported by Xiao et al. [26], Hua et al. [35], Gramaje et al. [22], and Zhang et al. [88]. However, Xangsayasane et al. [31] and Singh et al. [33] reported a significant association between GD, heterosis and hybrid performance. Our results also demonstrate that SCA effects were considerably related with hybrid performance and heterotic effects for all of the studied traits under water-stress and well-watered conditions. This indicates that SCA is a key factor in determining heterosis and the performance of the hybrids under water-deficit conditions. These results concur with the findings of Gramaje et al. [22], who reported that SCA estimates could be employed as a good predictor for hybrids’ performance and heterosis.

Information on the interrelationships between grain yield and other characteristics could reinforce the efficacy of breeding programs by allowing the exploitation of suitable traits as selection criteria under stress conditions [89,90,91]. The PC-biplot provides a suitable approach to assess such associations between studied traits [92,93]. Strong positive relationships were identified between grain yield and each of relative water content, chlorophyll content, number of panicles/plant, number of filled grains/panicle, and 1000-grain weight. This indicates their importance as valuable characteristics for indirect selection, particularly in segregated early generations, under drought stress [7,8,94]. In addition, grain yield was adversely associated with sterility percentage and leaf rolling [11,94,95].

## 5. Conclusions

Noticeable genetic variability was detected among the tested parents and their F_1_ hybrids for all the studied traits under well-watered and water-deficit conditions. The parental genotypes P_3_, P_4_, and P_6_ were identified as excellent general combiners for developing high-yielding and drought-tolerant rice genotypes. Moreover, the crosses P_1_ × P_4_, P_2_ × P_4_, P_3_ × P_4_, and P_4_ × P_6_ were the most favorable combinations for improving grain yield, particularly under water-deficit conditions. These are the best hybrid combinations with the highest better-parent and mid-parent heterosis for grain yield and other agronomic traits. Additive and non-additive gene effects are implicated in regulating the inheritance of all traits, with a prevalence of the additive gene action for most traits. Different plant traits comprising chlorophyll content, relative water content, number of panicles/plant, number of filled grains/panicle, and 1000-grain weight were recognized as indirect selection criteria for breeding drought-tolerant genotypes due to their positive association with grain yield. SSR based-genetic distance failed to predict SCA effects, heterosis, and hybrid performance. However, SCA could be employed in order to predict the heterosis and performance of hybrids under water-stress conditions.

## Figures and Tables

**Figure 1 plants-11-00702-f001:**
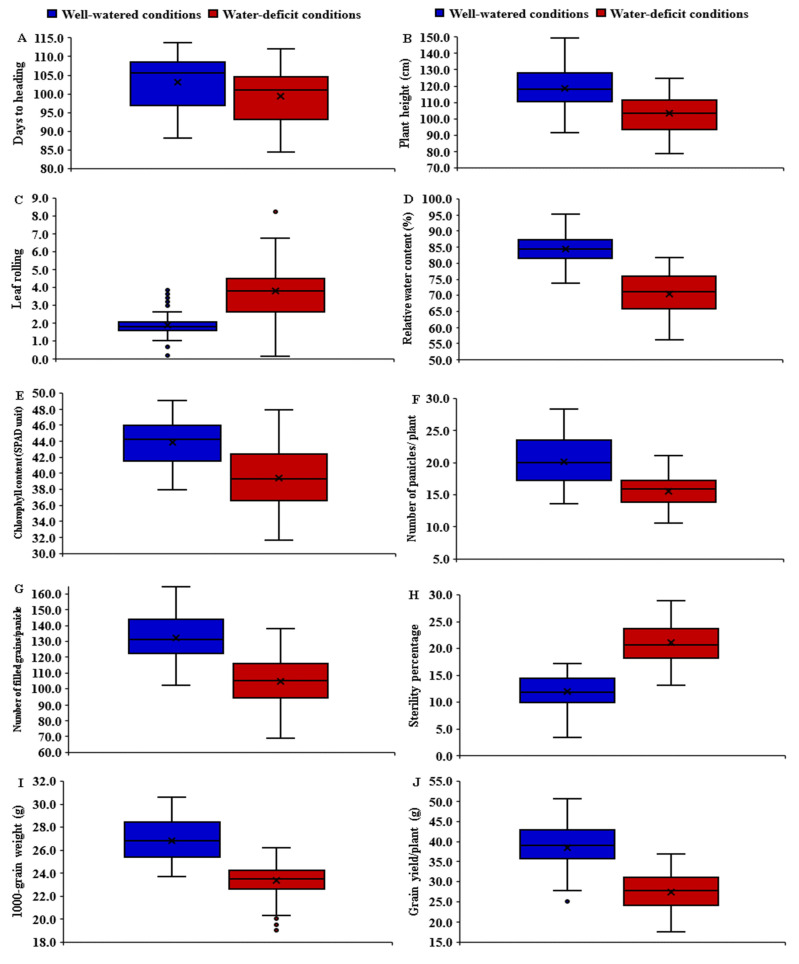
Boxplots for days to heading (**A**), plant height (**B**), leaf rolling (**C**), relative water content (**D**), chlorophyll content (**E**), number of panicles/plant (**F**), number of filled grains per panicle (**G**), sterility % (**H**), 1000-grain weight (**I**), and grain yield per plant (**J**) under well-watered and water-deficit conditions.

**Figure 2 plants-11-00702-f002:**
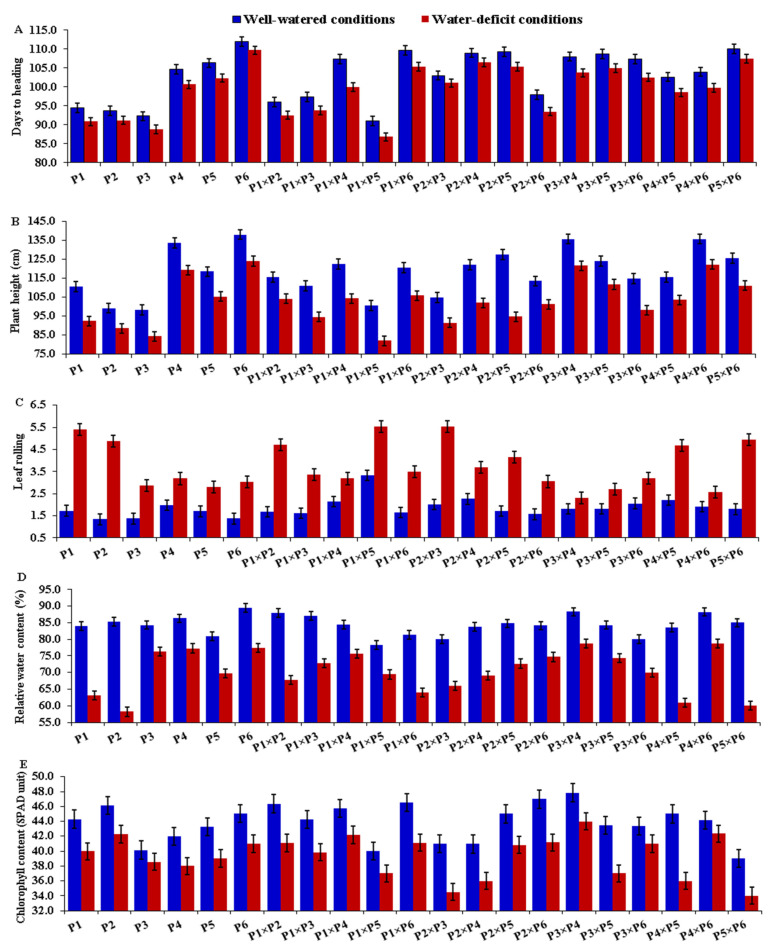
Mean performance of the six rice parents and their 15 F_1_ crosses for days to heading (**A**), plant height (**B**), leaf rolling (**C**), relative water content (**D**), and chlorophyll content (**E**) under well-watered and water-deficit conditions. The bars on the columns correspond to LSD (*p* ≤ 0.05).

**Figure 3 plants-11-00702-f003:**
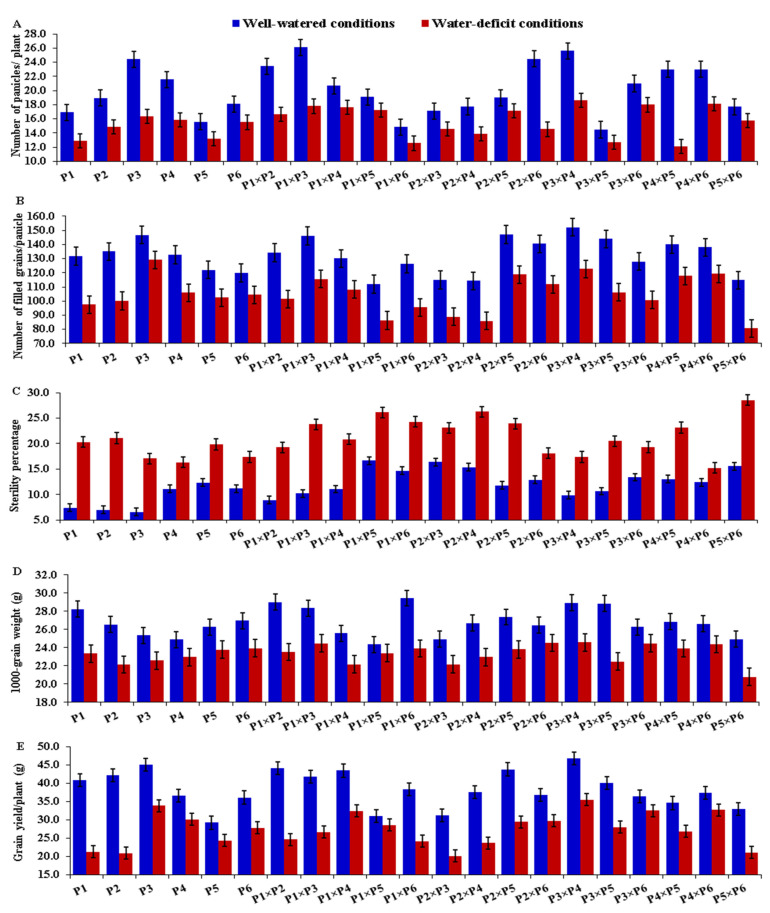
Mean performance of the six rice parents and their 15 F_1_ crosses for number of panicles per plant (**A**), number of filled grains per panicle (**B**), sterility % (**C**), 1000-grain weight (**D**), and grain yield per plant (**E**) under well-watered and water-deficit conditions. The bars on the columns represent LSD (*p* ≤ 0.05).

**Figure 4 plants-11-00702-f004:**
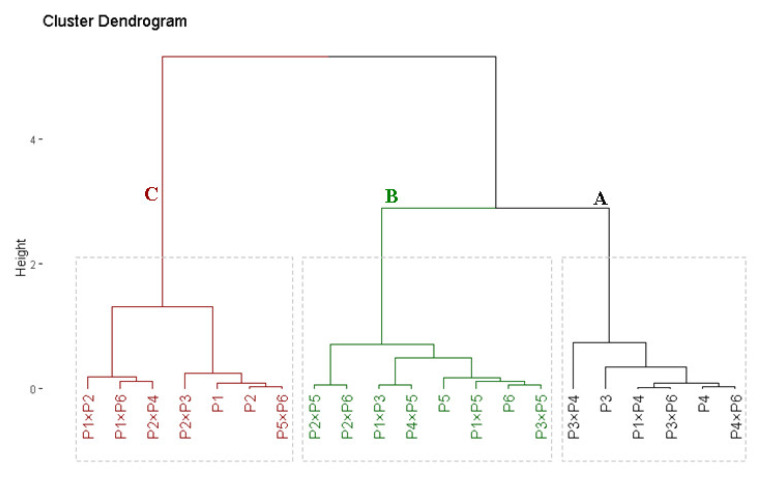
Dendrogram of the phenotypic distances among six rice genotypes and their 15 corresponding crosses based on four drought tolerance indices. The genotypes were categorized into three groups: A is drought-tolerant; B is moderately drought-tolerant; and C is drought-sensitive.

**Figure 5 plants-11-00702-f005:**
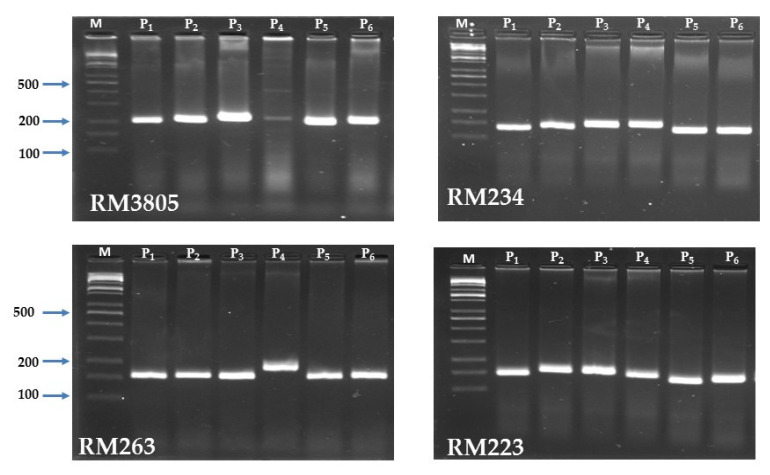
PCR amplified fragments for the SSR markers RM263, RM223, RM234, and RM3805 with the six parental rice genotypes. M is a 100 bp DNA ladder.

**Figure 6 plants-11-00702-f006:**
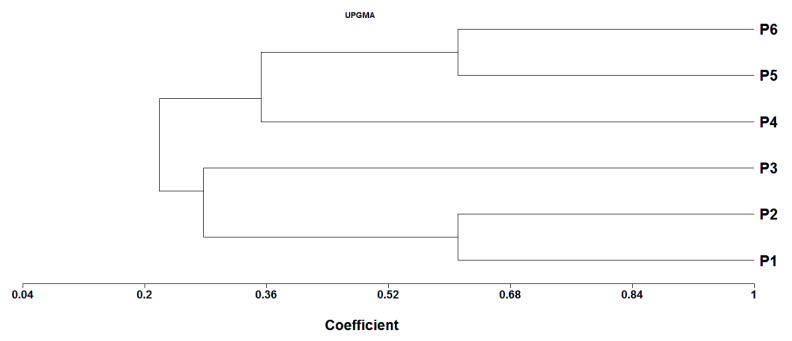
Dendrogram generated from UPGMA cluster analysis of the six parental rice genotypes based on SSR markers.

**Figure 7 plants-11-00702-f007:**
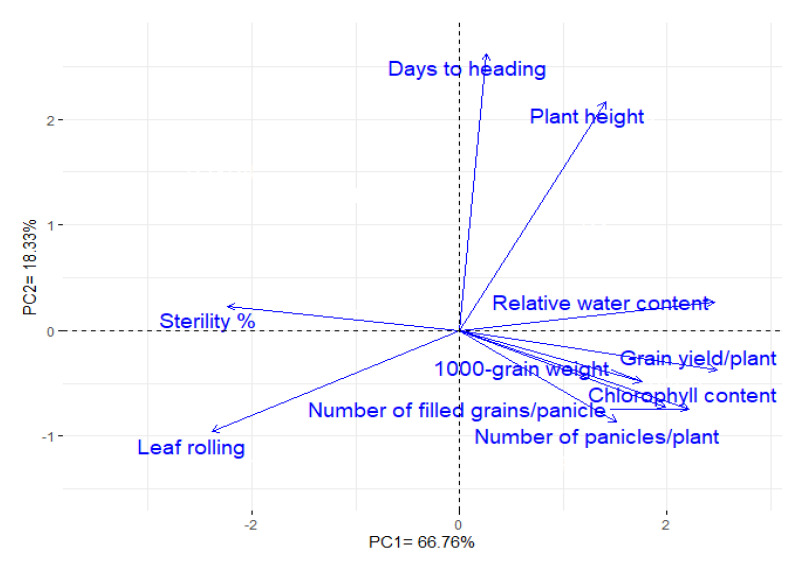
Biplot of principal component analysis exploring the association among the studied traits under water-deficit conditions.

**Table 1 plants-11-00702-t001:** Name, parentage and origin of the six rice genotypes utilized in the current study.

Code	Name	Parentage	Origin	Drought Tolerance
P_1_	Sakha-102	GZ4096/Giza177	Egypt	Sensitive
P_2_	Sakha-105	GZ5581-46-3/GZ 4316-7-1-1	Egypt	Sensitive
P_3_	Giza-179	GZ6296/GZ1368	Egypt	Tolerant
P_4_	Dullar	Unknown	India	Tolerant
P_5_	IRAT-112	(IRAT 13/dourado precoce)	Ivory Coast	Tolerant
P_6_	Moroberekan	IR 8-24-6-(M307 H5)	Guinea (West Africa)	Tolerant

**Table 2 plants-11-00702-t002:** Combined analysis of variance for all studied traits across tested environments.

Source of Variance	DF	DTH	PH	LR	RWC	CHLC	NP	NFG	SP	TGW	GYPP
Years (Y)	1	59.21 *	670.29 *	56.02 **	938.71 **	545.18 **	67.51 **	482.91 *	55.81 **	15.37 *	76.42 **
Replication/Y	4	6.23	60.96	0.4	4.06	9.5	1.12	41.18	0.89	1.22	2.97
Irrigation (I)	1	898.47 **	15047.07 **	231.23 **	12377.84 **	1250.43 **	1340.22 **	46572.20 **	5346.77 **	753.61 **	7674.90 **
Y × I	1	145.51 **	857.33 **	41.31 **	89.51 **	43.29*	7.99 *	681.84 *	11.60 *	6.45 *	31.21 **
Error a	4	6.18	29.51	0.33	3.67	5.32	0.92	35.17	0.73	0.84	2.36
Genotype (G)	20	527.42 **	1693.12 **	4.81 **	194.00 **	74.93 **	77.73 **	1801.61 **	101.16 **	14.24 **	190.07 **
G × Y	20	6.29 **	15.49 **	1.88 **	10.26 **	3.52 **	8.12 **	169.64 **	5.77 **	2.67 **	25.64 **
G × I	20	3.89 **	66.62 **	2.93 **	102.63 **	8.47 **	24.06 **	112.19 **	28.74 **	5.87 **	82.93 **
G × Y × I	20	6.00 **	26.80 **	1.09 **	8.12 **	5.32 **	3.57 **	77.00 **	4.08 **	1.26 *	19.53 **
Pooled Error	160	1.01	4.94	0.18	1.29	1.05	0.87	29.48	0.62	0.63	2.18
GCA	5	892.34 **	4309.89 **	6.97 **	284.41 **	57.28 **	87.79 **	1695.02 **	77.75 **	5.66 **	267.52 **
SCA	15	405.78 **	820.86 **	4.09 **	163.86 **	80.81 **	74.38 **	1837.15 **	108.96 **	17.10 **	164.25 **
GCA × Y	5	6.57 **	10.67	2.48 **	20.84 **	3.54 **	8.08 **	149.50 **	7.17 **	2.49 **	13.68 **
SCA × Y	15	6.20 **	17.09 **	1.68 **	6.73 **	3.51 **	8.13 **	176.35 **	5.30 **	2.72 **	29.63 **
GCA × I	5	4.44 **	20.53 **	5.94 **	128.00 **	4.75 **	17.43 **	120.99 **	53.53 **	3.67 **	188.29 **
SCA × I	15	3.71 **	81.99 **	1.93 **	94.17 **	9.71 **	26.27 **	109.26 **	20.48 **	6.60 **	47.81 **
GCA × Y × I	5	4.23 **	24.65 **	1.60 **	3.45 *	3.59 **	1.15	10.52	3.61 **	0.84	13.17 **
SCA × Y × I	15	6.60 **	27.51 **	0.92 **	9.68 **	5.89 **	4.37 **	99.16 **	4.24 **	1.40 **	21.65 **
GCA/SCA		2.2	5.25	1.7	1.74	0.71	1.18	0.92	0.71	0.33	1.63
GCAxY/SCAxY		1.06	0.62	1.47	3.1	1.01	0.99	0.85	1.35	0.92	0.46
GCAxI/SCAxI		1.19	0.25	3.07	1.36	0.49	0.66	1.11	2.61	0.56	3.94

* and ** indicate *p*-value < 0.05 and 0.01, respectively. DF is degree of freedom; DTH is days to heading; PH is plant height; CHLC is chlorophyll content (SPAD reading); LR is leaf rolling; RWC is relative water content; NP is number of panicles per plant; SP is sterility percentage; TGW is 1000-grain weight (g); and GYPP is grain yield per plant (g).

**Table 3 plants-11-00702-t003:** General combining ability estimates (GCA) of the six parents for all the studied traits under well-watered and water-deficit conditions.

**Parent**	**Days to Heading**		**Plant Height**		**Leaf Rolling**		**Relative Water Content**		**Chlorophyll Content**	
	**Well-** **Watered**	**Water-** **Deficit**	**Well-** **Watered**	**Water-** **Deficit**	**Well-** **Watered**	**Water-** **Deficit**	**Well-** **Watered**	**Water-** **Deficit**	**Well-** **Watered**	**Water-** **Deficit**
P_1_	−3.90 **	−4.36 **	−4.73 **	−5.66 **	0.11 *	0.58 **	−0.44 **	−2.04 **	0.58 **	0.69 **
P_2_	−2.37 **	−1.73 **	−5.90 **	−6.28 **	−0.14 **	0.56 **	0.10	−3.23 **	0.72 **	0.31 *
P_3_	−1.58 **	−1.42 **	−5.33 **	−4.34 **	−0.12 *	−0.45 **	−0.28	2.79 **	−0.84 **	−0.27 *
P_4_	2.34 **	1.84 **	8.71 **	8.96 **	0.16 **	−0.45 **	1.31 **	3.17 **	0.11	0.11
P_5_	1.59 **	1.57 **	0.13	−0.90 **	0.16 **	0.15 **	−1.59 **	−1.93 **	−0.98 **	−1.60 **
P_6_	3.92 **	4.09 **	7.13 **	8.22 **	−0.16 **	−0.39 **	0.90 **	1.24 **	0.41 **	0.75 **
LSD (gi) _0.05_	0.28	0.25	0.59	0.58	0.11	0.12	0.29	0.31	0.28	0.26
LSD (gi) _0.01_	0.37	0.33	0.78	0.76	0.14	0.15	0.38	0.41	0.37	0.35
**Parent**	**Number of** **Panicles/Plant**		**Number of** **Filled Grains/Panicle**		**Sterility Percentage**		**1000-Grain** **Weight**		**Grain Yield/Plant**	
	**Well-** **Watered**	**Water-** **Deficit**	**Well-** **Watered**	**Water-** **Deficit**	**Well-** **Watered**	**Water-** **Deficit**	**Well-** **Watered**	**Water-** **Deficit**	**Well-** **Watered**	**Water-** **Deficit**
P_1_	−0.37 **	−0.13	−1.45 *	−3.91 **	−0.82 **	0.95 **	0.69 **	0.09	1.44 **	−1.54 **
P_2_	−0.16	−0.26 *	−0.31	−3.25 **	−0.44 **	0.70 **	−0.01	−0.26 *	1.12 **	−2.76 **
P_3_	1.53 **	0.71 **	6.90 **	7.44 **	−1.14 **	−1.13 **	0.05	−0.01	2.22 **	2.38 **
P_4_	1.53 **	0.43 **	2.09 **	4.07 **	0.14	−1.47 **	−0.40 **	0.06	0.53 **	2.49 **
P_5_	−2.06 **	−0.92 **	−2.73 **	−2.33 **	1.20 **	1.82 **	−0.34 **	−0.18	−3.50 **	−1.08 **
P_6_	−0.47 **	0.17	−4.50 **	−2.03 **	1.05 **	−0.88 **	0.01	0.31 **	−1.82 **	0.52 **
LSD (gi) _0.05_	0.26	0.23	1.43	1.41	0.17	0.24	0.20	0.22	0.40	0.37
LSD (gi) _0.01_	0.35	0.30	1.90	1.88	0.22	0.32	0.27	0.29	0.53	0.50

* and ** indicate *p* < 0.05 and 0.01, respectively.

**Table 4 plants-11-00702-t004:** Specific combining ability effects (SCA) of 15 F_1_ cross combinations for all evaluated traits under well-watered and water-deficit conditions.

**Cross**	**Days to Heading**		**Plant Height**		**Leaf Rolling**		**Relative Water Content**		**Chlorophyll Content**	
	**Well-** **Watered**	**Water-** **Deficit**	**Well-** **Watered**	**Water-** **Deficit**	**Well-** **Watered**	**Water-** **Deficit**	**Well-** **Watered**	**Water-** **Deficit**	**Well-** **Watered**	**Water-** **Deficit**
P_1_ × P_2_	−0.83 *	−0.66	7.70 **	13.07 **	−0.15	−0.21	3.95 **	2.62 **	1.24 **	0.72 *
P_1_ × P_3_	−0.26	0.27	2.53 **	1.56	−0.23	−0.55 **	3.40 **	1.79 **	0.66	0.05
P_1_ × P_4_	5.82 **	3.19 **	0.11	−1.93 *	0.01	−0.73 **	−0.82 *	4.11 **	1.23 **	1.99 **
P_1_ × P_5_	−9.77 **	−9.69 **	−13.31 **	−14.40 **	1.21 **	1.03 **	−4.00 **	3.05 **	−3.42 **	−1.47 **
P_1_ × P_6_	6.54 **	6.28 **	−0.30	0.32	−0.15	−0.47 **	−3.50 **	−5.52 **	1.69 **	0.31
P_2_ × P_3_	3.87 **	4.92 **	−2.35 **	−0.84	0.41 **	1.64 **	−4.16 **	−3.88 **	−2.71 **	−4.91 **
P_2_ × P_4_	5.93 **	7.08 **	0.93	−3.64 **	0.38*	−0.18	−1.99 **	−1.27 **	−3.72 **	−3.80 **
P_2_ × P_5_	7.03 **	6.18 **	14.86 **	−1.07	−0.16	−0.33 *	1.92 **	7.50 **	1.44 **	2.74 **
P_2_ × P_6_	−6.65 **	−8.17 **	−6.14 **	−3.73 **	0.01	−0.89 **	−1.23 **	6.34 **	2.03 **	0.71
P_3_ × P_4_	4.15 **	4.03 **	13.86 **	14.05 **	−0.09	−0.58 **	2.92 **	2.49 **	4.72 **	4.79 **
P_3_ × P_5_	5.58 **	5.49 **	10.70 **	13.94 **	−0.09	−0.78 **	1.81 **	3.19 **	1.43 **	−0.50
P_3_ × P_6_	1.91 **	0.57	−5.53 **	−8.65 **	0.48 **	0.26	−4.96 **	−4.34 **	−0.06	1.15 **
P_4_ × P_5_	−4.36 **	−4.18 **	−11.75 **	−7.52 **	0.04	1.22 **	−0.47	−10.56 **	2.04 **	−1.89 **
P_4_ × P_6_	−5.37 **	−5.47 **	1.45	2.12 **	0.05	−0.35 *	1.70 **	4.03 **	−0.18	2.10 **
P_5_ × P_6_	1.42 **	2.45 **	−0.17	0.72	−0.05	1.42 **	1.35 **	−9.63 **	−4.26 **	−4.52 **
LSD Sij _0.05_	0.76	0.68	1.62	1.58	0.29	0.32	0.80	0.84	0.76	0.72
LSD Sij _0.01_	1.01	0.91	2.15	2.10	0.38	0.42	1.06	1.12	1.00	0.95
**Cross**	**Number of Panicles/Plant**		**Number of Filled Grains/Panicle**		**Sterility Percentage**		**1000-Grain Weight**		**Grain Yield/Plant**	
	**Well-** **Watered**	**Water-** **Deficit**	**Well-** **Watered**	**Water-** **Deficit**	**Well-** **Watered**	**Water-** **Deficit**	**Well-** **Watered**	**Water-** **Deficit**	**Well-** **Watered**	**Water-** **Deficit**
P_1_ × P_2_	3.83 **	1.50 **	3.97 *	3.98 *	−1.68 **	−3.42 **	1.52 **	0.37	3.13 **	1.59 **
P_1_ × P_3_	4.80 **	1.68 **	8.60 **	7.18 **	0.33	2.94 **	0.81 **	1.05 **	−0.21	−1.56 **
P_1_ × P_4_	−0.61	1.82 **	−2.51	3.20	−0.10	0.32	−1.52 **	−1.34 **	3.09 **	4.14 **
P_1_ × P_5_	1.40 **	2.78 **	−15.77 **	−12.22 **	4.45 **	2.30 **	−2.82 **	0.15	−5.36 **	3.90 **
P_1_ × P_6_	−4.47 **	−3.01 **	0.30	−3.19	2.60 **	3.17 **	1.94 **	0.15	0.29	−2.19 **
P_2_ × P_3_	−4.40 **	−1.38 **	−23.55 **	−19.96 **	6.11 **	2.51 **	−1.90 **	−0.90 **	−10.52 **	−6.84 **
P_2_ × P_4_	−3.77 **	−1.80 **	−19.40 **	−19.76 **	3.83 **	6.01 **	0.29	−0.17	−2.47 **	−3.44 **
P_2_ × P_5_	1.09 **	2.81 **	18.08 **	19.70 **	−0.79 **	0.36	0.92 **	0.92 **	7.79 **	5.96 **
P_2_ × P_6_	4.99 **	−0.90 **	13.35 **	12.26 **	0.47 *	−2.74 **	−0.34	1.12 **	−0.88	4.59 **
P_3_ × P_4_	2.43 **	1.96 **	11.31 **	6.52 **	−0.94 **	−1.07 **	2.48 **	1.19 **	5.71 **	3.30 **
P_3_ × P_5_	−5.13 **	−2.66 **	7.88 **	−3.57	−1.23 **	−1.25 **	2.34 **	−0.68*	2.90 **	−0.60
P_3_ × P_6_	−0.20	1.59 **	−6.35 **	−9.38 **	1.66 **	0.26	−0.60*	0.83 **	−2.32 **	2.23 **
P_4_ × P_5_	3.40 **	−2.89 **	8.69 **	11.29 **	−0.15	1.75 **	0.79 **	0.69 *	−0.82	−1.87 **
P_4_ × P_6_	1.80 **	2.03 **	8.51 **	12.52 **	−0.62 **	−3.48 **	0.22	0.65 *	0.28	2.36 **
P_5_ × P_6_	0.09	0.96 **	−9.89 **	−19.79 **	1.48 **	6.54 **	−1.53 **	−2.70 **	−0.12	−5.74 **
LSD Sij _0.05_	0.72	0.63	3.94	3.88	0.46	0.66	0.55	0.59	1.10	1.03
LSD Sij _0.01_	0.95	0.83	5.22	5.15	0.60	0.87	0.73	0.79	1.46	1.36

* and ** indicate *p*-value < 0.05 and 0.01, respectively.

**Table 5 plants-11-00702-t005:** Heterosis relative to Mid-Parent (MP) and Better-Parent (BP) of the 15 F_1_ hybrids for all evaluated traits under well-watered (W) and water-deficit (S) conditions.

**Cross**	**Days to Heading**				**Plant Height**				**Leaf Rolling**				**Relative Water Content (%)**				**Chlorophyll Content**			
	**M.P**		**B.P**		**M.P**		**B.P**		**M.P**		**B.P**		**M.P**		**B.P**		**M.P**		**B.P**	
	**W**	**S**	**W**	**S**	**W**	**S**	**W**	**S**	**W**	**S**	**W**	**S**	**W**	**S**	**W**	**S**	**W**	**S**	**W**	**S**
P_1_ × P_2_	1.99 **	1.67 **	2.47 **	1.83 **	10.14 **	15.12 **	16.51 **	17.61 **	9.47	−8.53	25.44	−3.62	3.93 **	11.47 **	3.12 **	7.05 **	2.55 *	−0.04	0.52	−2.76 *
P_1_ × P_3_	4.21 **	4.39 **	5.47 **	5.60 **	6.23 **	7.07 **	12.95 **	12.20 **	3.66	−18.76 **	16.99	16.78	3.46 **	4.47 **	3.26 **	−4.47 **	4.77 **	1.50	−0.14	−0.30
P_1_ × P_4_	7.77 **	4.39 **	13.55 **	10.01 **	0.36	−1.42	10.78 **	13.00 **	15.34	−26.02 **	23.75	−0.26	−0.86	7.58 **	−2.21 **	−2.20 *	6.04 **	8.17 **	3.30 *	5.52 **
P_1_ × P_5_	−9.39 **	−10.12 **	−3.74 **	−4.46 **	−12.26 **	−17.03 **	−9.12 **	−11.19 **	94.74 **	31.61 **	96.46 **	98.44 **	−4.96 **	4.43 **	−6.71 **	−0.42	−8.61 **	−6.28 **	−9.69 **	−7.41 **
P_1_ × P_6_	6.17 **	5.03 **	15.98 **	15.88 **	−3.05 **	−2.15 *	8.97 **	14.63 **	6.62	−16.82 **	20.58	16.09	−6.24 **	−8.95 **	−9.13 **	−17.26 **	4.11 **	1.58	3.25*	0.29
P_2_ × P_3_	10.77 **	12.31 **	11.60 **	13.79 **	6.28 **	5.93 **	6.79 **	8.61 **	47.97 **	42.59 **	50.00 **	92.01 **	−5.64 **	−1.84 *	−6.20 **	−13.47 **	−4.93 **	−14.62 **	−11.10 **	−18.38 **
P_2_ × P_4_	9.89 **	11.03 **	16.35 **	16.81 **	5.01 **	−1.84	23.24 **	15.25 **	36.35 **	−8.19	69.08 **	16.23 *	−2.37 **	1.86 *	−2.94 **	−10.70 **	−7.08 **	−10.30 **	−11.24 **	−14.83 **
P_2_ × P_5_	9.34 **	8.87 **	16.72 **	15.54 **	17.18 **	−2.28 *	28.64 **	7.02 **	13.63	5.33	28.93	48.92 **	2.04 **	13.64 **	−0.61	4.28 **	0.70	0.49	−2.43	−3.39 *
P_2_ × P_6_	−4.72 **	−6.89 **	4.61 **	2.56 **	−4.27 **	−4.78 **	14.52 **	14.31 **	16.08	−22.70 **	17.46	1.16	−3.74 **	10.16 **	−5.99 **	−3.47 **	3.07 **	−1.19	1.86	−2.67
P_3_ × P_4_	9.65 **	9.51 **	17.01 **	16.81 **	17.09 **	19.50 **	38.19 **	44.37 **	7.81	−4.34	31.55	−20.14 *	3.53 **	2.60 **	2.33 **	1.93 *	16.46 **	14.97 **	13.87 **	14.15 **
P_3_ × P_5_	9.43 **	9.80 **	17.74 **	18.15 **	14.38 **	17.80 **	26.23 **	32.54 **	17.70	−8.22	31.55	−3.11	2.07 **	1.90 *	0.00	−2.49 **	4.20 **	−4.57 **	0.45	−5.13 **
P_3_ × P_6_	5.07 **	3.32 **	16.29 **	15.45 **	−2.89 **	−5.70 **	16.82 **	16.55 **	50.03 **	8.57	50.30 **	11.11	−7.91 **	−8.87 **	−10.58 **	−9.51 **	1.78	3.08 *	−3.76 **	0.00
P_4_ × P_5_	−2.70 **	−2.91 **	−1.94 **	−2.11 **	−8.35 **	−7.86 **	−2.53 *	−1.74	20.45	51.72 **	30.45 *	68.42 **	0.02	−16.98 **	−3.13 **	−21.06 **	5.57 **	−6.49 **	4.04 **	−7.69 **
P_4_ × P_6_	−4.03 **	−5.13 **	−0.67	−0.88	−0.06	0.48	1.62	2.45 *	13.94	−3.79	39.34 *	−14.32	0.42	1.87 *	−1.37	1.81 *	1.49	7.18 **	−1.94	3.26 *
P_5_ × P_6_	0.78	1.35 **	3.48 **	4.98 **	−2.15*	−3.20 **	5.91 **	5.38 **	17.67	64.50 **	31.79	77.63 **	−0.19	−18.15 **	−4.99 **	−22.44 **	−11.65 **	−15.00 **	−13.41 **	−17.07 **
**Cross**	**Number of Panicles/Plant**				**Number of Filled Grains/Panicle**				**Sterility Percentage**				**1000-Grain Weight**				**Grain Yield/Plant**			
	**M.P**		**B.P**		**M.P**		**B.P**		**M.P**		**B.P**		**M.P**		**B.P**		**M.P**		**B.P**	
	**W**	**S**	**W**	**S**	**W**	**S**	**W**	**S**	**W**	**S**	**W**	**S**	**W**	**S**	**W**	**S**	**W**	**S**	**W**	**S**
P_1_ × P_2_	30.69 **	19.76 **	23.77 **	11.74 **	0.55	2.80	−0.70	1.43	23.76 **	−6.82 **	27.28 **	−4.99	5.92 **	3.51	2.76	0.89	6.34 **	16.71 **	4.69 *	15.53 **
P_1_ × P_3_	26.19 **	21.67 **	6.81 **	8.78 **	4.83 *	1.87	−0.54	−10.65 **	46.05 **	27.40 **	54.96 **	39.39 **	5.88 **	6.60 **	0.45	4.86 *	−2.54	−3.45	−7.14 **	−21.31 **
P_1_ × P_4_	7.45 **	22.91 **	−4.13	11.47 **	−1.67	6.27 *	−2.06	1.99	19.57 **	13.85 **	49.62 **	27.59 **	−3.64 *	−4.28 *	−9.35 **	−5.04 *	12.33 **	26.18 **	6.49 **	7.83 **
P_1_ × P_5_	17.48 **	32.37 **	12.84 **	30.91 **	−11.73 **	−13.64 **	−15.00 **	−15.72 **	68.86 **	30.20 **	125.61 **	31.69 **	−10.67 **	−0.69	−13.77 **	−1.64	−11.51 **	25.27 **	−24.08 **	17.47 **
P_1_ × P_6_	−15.29 **	−11.65 **	−18.03 **	−19.15 **	0.34	−5.30	−4.14	−8.48 **	58.36 **	28.89 **	98.53 **	39.59 **	6.70 **	1.11	4.33 **	−0.16	−0.40	−1.66	−6.12 **	−13.07 **
P_2_ × P_3_	−21.13 **	−6.57 *	−30.00 **	−10.77 **	−18.41 **	−22.41 **	−21.66 **	−31.14 **	141.16 **	21.17 **	148.58 **	35.41 **	−3.85 *	−0.88	−6.03 **	−1.82	−28.36 **	−26.40 **	−30.70 **	−40.49 **
P_2_ × P_4_	−12.43 **	−9.54 **	−17.78 **	−12.26 **	−14.65 **	−16.74 **	−15.37 **	−19.04 **	69.90 **	40.46 **	119.87 **	60.85 **	3.81 *	1.84	0.55	0.04	−4.48 *	−7.24 *	−10.78 **	−21.40 **
P_2_ × P_5_	10.07 **	22.21 **	0.35	15.21 **	14.35 **	17.39 **	8.80 **	16.10 **	22.15 **	16.94 **	69.02 **	20.63 **	3.65 *	3.70 *	3.12	0.13	22.82 **	30.22 **	4.01	20.95 **
P_2_ × P_6_	32.33 **	−4.42	29.40 **	−6.40	10.15 **	9.21 **	3.99	6.94 *	42.60 **	−5.83 *	84.90 **	4.16	−1.09	6.40 **	−1.87	2.43	−5.87 **	22.07 **	−12.57 **	6.96 *
P_3_ × P_4_	11.38 **	15.73 **	4.87 *	13.89 **	8.90 **	4.39	3.72	−4.99 *	11.92 **	3.94	50.30 **	6.27	15.27 **	7.96 **	14.23 **	7.05 **	14.80 **	11.10 **	3.97 *	4.91 *
P_3_ × P_5_	−27.69 **	−14.29 **	−40.78 **	−22.61 **	7.14 **	−8.22 **	−1.91	−17.75 **	12.65 **	10.94 **	62.00 **	19.91 **	11.84 **	−3.09	9.85 **	−5.56 **	7.74 **	−3.62	−11.20 **	−17.13 **
P_3_ × P_6_	−1.25	12.86 **	−14.05 **	10.00 **	−4.04	−13.75 **	−12.81 **	−22.01 **	51.35 **	11.88 **	103.67 **	12.94 **	0.43	5.26 **	−2.61	2.26	−10.14 **	5.42 *	−19.06 **	−4.05
P_4_ × P_5_	23.78 **	−16.30 **	6.61 *	−23.33 **	9.88 **	13.04 **	5.39 *	11.11 **	11.09 **	27.95 **	17.36 **	41.62 **	5.02 **	2.32	2.25	0.55	5.22 *	−1.23	−5.35 *	−10.61 **
P_4_ × P_6_	15.96 **	15.75 **	6.61 *	14.62 **	9.21 **	13.36 **	3.93	12.55 **	11.59 **	−9.80 **	11.77 **	−6.88 *	2.72	3.95 *	−1.26	1.83	2.88	13.10 **	2.25	8.76 **
P_5_ × P_6_	5.04	9.58 **	−2.24	1.30	−5.10 *	−22.08 **	−5.88 *	−22.86 **	32.51 **	53.20 **	39.75 **	63.92 **	−6.30 **	−12.94 **	−7.52 **	−13.21 **	0.88	−19.21 **	−8.77 **	−24.16 **

* and ** indicate *p*-value < 0.05 and 0.01, respectively.

**Table 6 plants-11-00702-t006:** Genetic details of the sixteen polymorphic SSR markers used in this study.

Marker Name	Chromosome Number	Number of Alleles	Effective Number of Alleles	Major Allele Frequency	Gene Diversity	PIC
RM315	1	2	1.95	0.58	0.49	0.37
RM543	1	2	1.80	0.67	0.44	0.35
RM263	2	3	2.32	0.58	0.57	0.50
RM279	2	4	3.79	0.33	0.74	0.69
RM55	3	3	2.0	0.67	0.50	0.45
RM518	4	4	3.60	0.33	0.72	0.67
RM159	5	4	3.0	0.50	0.67	0.62
RM3805	6	3	2.57	0.50	0.61	0.54
RM70	7	3	2.57	0.50	0.61	0.54
RM234	7	3	2.57	0.50	0.61	0.54
RM72	8	4	3.60	0.33	0.72	0.67
RM223	8	4	3.60	0.33	0.72	0.67
RM160	9	3	2.57	0.50	0.61	0.54
RM222	10	3	2.57	0.50	0.61	0.54
RM332	11	3	2.0	0.67	0.50	0.45
RM20A	12	5.	3.79	0.33	0.74	0.69
**Mean**		3.31	2.77	0.49	0.62	0.55

**Table 7 plants-11-00702-t007:** Genetic distance among the assessed parental genotypes based on SSR markers.

Parent	P_1_	P_2_	P_3_	P_4_	P_5_	P_6_
P_1_	-					
P_2_	0.38	-				
P_3_	0.78	0.67	-			
P_4_	0.88	0.82	0.77	-		
P_5_	0.89	0.72	0.78	0.59	-	
P_6_	0.78	0.61	0.78	0.71	0.39	-

**Table 8 plants-11-00702-t008:** Correlation coefficients among parental genetic distance (GD), F_1_ hybrid performance, mid-parent heterosis (MP), better-parent (BP) heterosis, and SCA for all the studied traits under well-watered (W) and water-deficit (S) conditions.

Correlation	Irrig.	Days to Heading	Plant Height	Leaf Rolling	Relative Water Content	Chlorophyll Content	Number of Panicles/Plant	Number of Filled Grains/Panicle	Sterility Percentage	1000-Grain Weight	Grain Yield/Plant
**r (GD, F_1_)**	**W**	0.08	−0.05	0.43	−0.35	0.02	−0.16	0.01	0.10	−0.04	0.08
	**S**	−0.01	−0.23	−0.36	0.49	0.24	0.16	0.11	−0.03	0.25	0.45
**r (GD, SCA)**	**W**	0.19	−0.06	0.38	−0.39	0.07	−0.33	−0.12	0.26	−0.07	−0.06
	**S**	0.12	−0.29	−0.33	0.34	0.27	0.07	−0.02	0.08	0.23	0.31
**r (GD, MP)**	**W**	0.22	−0.03	0.34	−0.29	0.19	−0.33	−0.13	0.14	0.03	−0.03
	**S**	0.15	−0.18	−0.42	0.25	0.35	0.13	−0.01	0.04	0.24	0.26
**r (GD, BP)**	**W**	0.40	0.09	0.38	−0.15	0.13	−0.35	−0.22	0.44	−0.05	−0.14
	**S**	0.34	0.07	−0.14	0.20	0.34	0.15	−0.11	0.05	0.29	0.09
**r (F_1_, SCA)**	**W**	0.80 **	0.65 **	0.91 **	0.92 **	0.95 **	0.90 **	0.93 **	0.88 **	0.96 **	0.84 **
	**S**	0.81 **	0.66 **	0.83 **	0.84 **	0.95 **	0.95 **	0.92 **	0.91 **	0.98 **	0.82 **
**r (F_1_, MP)**	**V**	0.68 **	0.53 *	0.88 **	0.86 **	0.89 **	0.84 **	0.91 **	0.68 **	0.91 **	0.77 **
	**S**	0.69 **	0.52 *	0.72 **	0.69	0.96 **	0.91 **	0.89 **	0.94 **	0.94 **	0.70 **
**r (F_1_, BP)**	**W**	0.73 **	0.53 *	0.89 **	0.83 **	0.92 **	0.77 **	0.84 **	0.66 **	0.87 **	0.80 **
	**S**	0.75 **	0.55 *	0.90 **	0.78 **	0.93 **	0.92 **	0.75 **	0.91 **	0.92	0.66 **
**r (SCA, MP)**	**W**	0.95 **	0.95 **	0.99 **	0.97 **	0.96 **	0.98 **	0.99 **	0.92 **	0.98 **	0.97 **
	**S**	0.96 **	0.97 **	0.95 **	0.95 **	0.95 **	0.97 **	0.98 **	0.98 **	0.97 **	0.97 **
**r (SCA, BP)**	**W**	0.89 **	0.83 **	0.94 **	0.91 **	0.97 **	0.92 **	0.96 **	0.86 **	0.96 **	0.88 **
	**S**	0.88 **	0.77 **	0.88 **	0.91 **	0.95 **	0.97 **	0.92 **	0.97 **	0.93 **	0.93 **

* and ** indicate *p*-value < 0.05 and 0.01, respectively.

## Data Availability

The data presented in this study are available upon request from the corresponding author.

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
