# Peer review of "Molecular Genetic Diversity and Combining Ability for Some Physiological and Agronomic Traits in Rice under Well-Watered and Water-Deficit Conditions"

_plants, 2022, doi:10.3390/plants11050702_

Round 1

Reviewer 1 Report

While the data and experimental design suggest that the manuscript contains valuable information its very poor English with lots of incomplete sentences and lacking sequence of tenses make the reading extremely difficult, with some hypotheses unclear/improperly defined. Needs major revision before any actual evaluation of the research itself!

Author Response

Re: We would like to thank the Reviewer for his time dedicated to our manuscript. The manuscript has been carefully reviewed and has been considerably improved.

Reviewer 2 Report

The paper by Sakran et al focuses on the study of genetic diversity and the identification of the most favorable combinations for improving rice yield under water stress conditions.

This is a subject with extreme importance in light of the climate change events we are experience and has interesting results, with practical use and applicability in breeding programs.

Minor issues are listed below.

L22-25: weren't these results expected? I believe the original findings here are related to the differences between genotypes.

L48-50: I don't agree with this sentence; current rice agricultural intensive monocrop growth is extremely detrimental to the environment and, if we aim at increasing its production by 60%, then agricultural practices could get even worse. I agree that it is important to find genotypes more resistant to climate change and understanding the genetic diversity is key to improving current agricultural practices.

L57-59: The impact of drought on the world's arable areas should be more supported to explain why drought tolerant genotypes can maintain global food security. Particularly from the point of view of rice.

L58: Developing shouldn't be in capitalised letter.

L86: while describing the objectives of the study is useful, it is very important to explicitly show the main hypothesis behind the study.

L119: change 4 to four

L121: change twelve to 12

L122: change fifteen to 15

L151,152: change ten to 10

L153: Add The 1000-grains...

L159: what were the criteria for the leaves to be considered 'healthful'? Did you use the SPAD measurements as quality control? Or any other methods? Like this it seems a little bit arbitrary.

L161: change sixteen to 16

Figure 9 has poor quality and could be removed.

L449: Change to 'In light of that'

L455: change to 'a pivotal'

L459: delete duplicated with

L468: change 'be resulted' with 'result'

L482: change 'The efficient' with 'An efficient'

Author Response

The paper by Sakran et al focuses on the study of genetic diversity and the identification of the most favorable combinations for improving rice yield under water stress conditions. This is a subject with extreme importance in light of the climate change events we are experienced and has interesting results, with practical use and applicability in breeding programs.

Re: We would like to thank the Reviewer for his time devoted to our manuscript, and his positive assessment of our work.

Minor issues are listed below.

L22-25: weren't these results expected? I believe the original findings here are related to the differences between genotypes.

Re: You are right, it was expected the negative effect of water deficit conditions on rice traits, but it was important to present that impact on the measured traits. 

L48-50: I don't agree with this sentence; current rice agricultural intensive mono-crop growth is extremely detrimental to the environment and, if we aim at increasing its production by 60%, then agricultural practices could get even worse. I agree that it is important to find genotypes more resistant to climate change and understanding the genetic diversity is key to improving current agricultural practices.

Re: The paragraph has been revised ad improved as suggested, please see lines 53-69 in the revised version.

L57-59: The impact of drought on the world's arable areas should be more supported to explain why drought tolerant genotypes can maintain global food security. Particularly from the point of view of rice.

Re: The impact of drought stress on rice and the importance of developing tolerant genotypes in maintaining global food security have been explained, please see lines 57-69.

L58: Developing shouldn't be in capitalised letter.

Re: The word has been modified (line 67)

L86: while describing the objectives of the study is useful, it is very important to explicitly show the main hypothesis behind the study.

Re: The hypothesis has been clarified, please see lines 95-97

L119: change 4 to four

Re: The number has been changed as suggested (line 131)

L121: change twelve to 12

Re: The number has been changed as suggested (line 134)

L122: change fifteen to 15

Re: The number has been changed as suggested (line 135)

L151,152: change ten to 10

Re: The number has been changed as suggested (line 159, 160, 163)

L153: Add The 1000-grains...

Re: “The” has been added before “1000-grains weight”  (line 161)

L159: what were the criteria for the leaves to be considered 'healthful'? Did you use the SPAD measurements as quality control? Or any other methods? Like this it seems a little bit arbitrary.

Re: The DNA was extracted from 20-day-old seedlings, the selected leave portion for extraction was healthful.

L161: change sixteen to 16

Re: The number has been changed as suggested (line 169)

Figure 9 has poor quality and could be removed.

Re: Figure 9 has been removed as suggested

L449: Change to 'In light of that'

Re: It has been modified as suggested (line 448)

L455: change to 'a pivotal'

Re: It has been modified as suggested (line 455)

L459: delete duplicated with

Re: The duplicated with has been deleted (line 458)

L468: change 'be resulted' with 'result'

Re: It has been modified as suggested (line 467)

L482: change 'The efficient' with 'An efficient'

Re: It has been modified as suggested (line 481)

Reviewer 3 Report

The manuscript title “Genetic diversity and combining ability of physiological and agronomic traits under well-watered and water deficit conditions in arid environment” presents novel and significant findings. The manuscript is perfectly planned and executed. I would like to recommend accepting this manuscript after addressing some minor changes from the authors, which will further enhance the quality of the work.

  1. Results section has lot of text, figures, and tables, I would recommend authors to concise things, present most important message there despite discussing everything.
  2. Please proofread manuscript for various typos
  3. Please provide the precipitation data for both years
  4. Line 178-179: How this issue can be resolved by combined data analysis

Author Response

The manuscript title “Genetic diversity and combining ability of physiological and agronomic traits under well-watered and water deficit conditions in arid environment” presents novel and significant findings. The manuscript is perfectly planned and executed. I would like to recommend accepting this manuscript after addressing some minor changes from the authors, which will further enhance the quality of the work.

Re: We would like to thank the Reviewer for his time dedicated to our manuscript, and his positive assessment of our work.

The results section has a lot of text, figures, and tables, I would recommend authors to concise things, present the most important message there despite discussing everything.

Re: The results section has been revised and two figures have been transferred to the supplementary materials.

Please proofread manuscript for various typos

Re: The manuscript has been carefully reviewed and typos have been corrected.

Please provide the precipitation data for both years

Re: More details have been added please see lines 119-120 in the revised version

Line 178-179: How this issue can be resolved by combined data analysis

Re: The tests for homogeneity of the variances revealed homogenous variances across the two growing seasons for the measured traits; consequently, the data of the two growing seasons were combined. More details have been added in lines 187-190.

Reviewer 4 Report

L.20. What is SCA? The decryption should be indicated at the first mention in the text.

The origin of SSR markers should be specified in Materials and Methods. Are they related to drought tolerance?

Tables S1 and S2 are missing in the manuscript.

Figures 3 and 4. Significance of differences between genotypes should be indicated.

Author Response

Reviewer 4:

L.20. What is SCA? The decryption should be indicated at the first mention in the text.

Re: It is specific combining ability, has been indicated as well as all abbreviations

The origin of SSR markers should be specified in Materials and Methods. Are they related to drought tolerance?

Re: Yes, the applied SSR markers were selected based on their relevance to drought tolerance. More details and references have been added in lines 168-169.

Tables S1 and S2 are missing in the manuscript.

Re: The supplementary materials are presented in a separate file

Figures 3 and 4. The significance of differences between genotypes should be indicated.

Re: The bars on the columns represent the least significant difference (LSD) at p ≤ 0.05, has been added in lines 273, 276.

Round 2

Reviewer 1 Report

The manuscript by Sakran et al. deals with an important task of improving rice yield under water stress conditions. The study takes an integrated approach, thus producing interesting results, with practical use and applicability in breeding programs.

While the revised version is significantly improved, there are still a few important things to be corrected:

  1. The statement that “diallel mating design” is an “analysis” (Lines 74-75) is incorrect as the “design” applies to the set-up of the experiment, while the “analysis” treats the data obtained from the experiment.
  2. I doubt that all 16 SSR primer pairs have an optimal annealing temperature of 55 °C! Therefore, the description of the PCR reaction conditions needs to be modified to reflect the correct ones.
  3. Throughout the text strange/unusual word usage abounds, which often makes the reader wonder what is authors' actual opinion on the matters discussed. Therefore, yet another revision of the usage of proper English language and scientific terms in particular contexts is needed, Just a few examples:
    • “the ability of SSR markers … is indecisive” (Lines 94-95)
    • “panicles were fully exerted in each plot” (Line 146)
    • “hybrids reflected noticeable variation” (Line 317)
    • “the crosses … recorded significant and negative SCA values” (Lines 319-320), etc.

Author Response

Dear Editor,

We would like to thank you and the reviewers for the time and efforts devoted to our manuscript. We have revised the manuscript according to the additional comments pointed out by the reviewer. We have addressed the comments of the reviewers in a point-by-point below in red color, in addition, we have highlighted all the associated changes made to the manuscript using track changes.

Yours sincerely,

Authors

Responses to Reviewer Comments

The manuscript by Sakran et al. deals with an important task of improving rice yield under water stress conditions. The study takes an integrated approach, thus producing interesting results, with practical use and applicability in breeding programs.

While the revised version is significantly improved, there are still a few important things to be corrected:

The statement that “diallel mating design” is an “analysis” (Lines 74-75) is incorrect as the “design” applies to the set-up of the experiment, while the “analysis” treats the data obtained from the experiment.

Re: We would like to thank the Reviewer for his time dedicated to our manuscript. The diallel cross is a mating scheme applied by plant breeders. It is common to be used as “diallel mating design” however as the reviewer suggested it has been replaced by “diallel mating analysis” please see lines 74-75 in the revised version.

I doubt that all 16 SSR primer pairs have an optimal annealing temperature of 55 °C! Therefore, the description of the PCR reaction conditions needs to be modified to reflect the correct ones.

Re: Thanks so much for your accuracy, more details have been added as suggested, please see lines 185-189.

Throughout the text strange/unusual word usage abounds, which often makes the reader wonder what is authors' actual opinion on the matters discussed. Therefore, yet another revision of the usage of proper English language and scientific terms in particular contexts is needed, Just a few examples: “the ability of SSR markers … is indecisive” (Lines 94-95)

“panicles were fully exerted in each plot” (Line 146). “hybrids reflected noticeable variation” (Line 317). “the crosses … recorded significant and negative SCA values” (Lines 319-320), etc.

Re: All manuscript has been revised and its language has been improved
